# Maternal Obesity Programs Offspring Development and Resveratrol Potentially Reprograms the Effects of Maternal Obesity

**DOI:** 10.3390/ijerph17051610

**Published:** 2020-03-02

**Authors:** Mei-Hsin Hsu, Yu-Chieh Chen, Jiunn-Ming Sheen, Li-Tung Huang

**Affiliations:** 1Department of Pediatrics, Kaohsiung Chang Gung Memorial Hospital and Chang Gung University College of Medicine, Kaohsiung 833, Taiwangesicht27@cgmh.org.tw (Y.-C.C.); e5724@cgmh.org.tw (J.-M.S.); 2Department of Medicine, Chang Gung University, Linkou 333, Taiwan

**Keywords:** maternal obesity, resveratrol, programming, reprogramming

## Abstract

Maternal obesity during pregnancy is a now a public health burden that may be the culprit underlying the ever-increasing rates of adult obesity worldwide. Understanding the association between maternal obesity and adult offspring’s obesity would inform policy and practice regarding offspring health through available resources and interventions. This review first summarizes the programming effects of maternal obesity and discusses the possible underlying mechanisms. We then summarize the current evidence suggesting that maternal consumption of resveratrol is helpful in maternal obesity and alleviates its consequences. In conclusion, maternal obesity can program offspring development in an adverse way. Maternal resveratrol could be considered as a potential regimen in reprogramming adverse outcomes in the context of maternal obesity.

## 1. Introduction

Obesity is a common metabolic disorder and is prevalent worldwide [1]. According to the World Health Organization, nearly 39% of adults aged 18 years and over were overweight in 2016, and 13% were obese [2]. In the United States, the prevalence of obese women aged 20–39 years old tripled between 1960 and 2000 [3]. Likewise, in the UK, between 1990 and 2004, the body mass index (BMI) of women with singleton pregnancies at their first prenatal booking increased by an average of 1.37 kg/m^2^ and the prevalence of obesity tripled [4]. Based on the US Institute of Medicine guidelines, approximately 40% of women gain an excessive amount of weight during pregnancy in Western countries [5]. Maternal obesity is also linked with individual’s socioeconomic position and environmental determinants of health [6]. Maternal obesity during pregnancy includes pre-pregnancy obesity and excessive gestational weight gain. Both pre-pregnancy maternal obesity and excessive gestational weight gain increase the placental transfer of nutrients to the developing fetus and affect fetal development. Maternal obesity during pregnancy represents significant health risks to both the mother and the baby, and is associated with an increased risk of miscarriage, gestational diabetes, stillbirth, preeclampsia leading to preterm birth, and Cesarean delivery [7]. The works cited in this review were identified eligible studies and reviews through manual searches related to maternal obesity. It is well-known that pre-pregnancy maternal obesity and excessive gestational weight gain have different mechanisms and clinical implications. In this manuscript, we use the term maternal obesity to mean maternal obesity during pregnancy.

## 2. Long-Term Public Health Issues of Maternal Obesity

Epidemiological and experimental studies provide evidence of a persistent and deleterious effect of maternal obesity on their offspring, supporting the hypothesis of intrauterine programming [8]. Clinical and animal studies have shown a maternal obesogenic environment, i.e., during the periods of periconception, pregnancy, or lactation, increasing the risk for the development of obesity and related cardiometabolic [9,10] and neurodevelopmental disorders [11] in offspring in adulthood. The rise in obesity prevalence is partly due to increase in pre-existing type 2 diabetes among pregnant women. Besides, maternal obesity is strongly linked with gestational diabetes [12]. Maternal obesity is a growing problem in both the developed and developing world because it can lead to a cyclical transgenerational transmission of obesity. Understanding the associations between maternal obesity and offspring health would inform public health policy, practice, and early intervention.

## 3. Specific Mechanisms Underlying How Maternal Obesity Programs Offspring

Growing evidence suggests that developmental programming is noted in obese mothers’ offspring [13]. Below are some major underlying mechanisms of how maternal obesity exerts its effects on the placenta, fetus, and offspring development.

### 3.1. Epigenome

Epigenetics is defined as mechanisms of long-term stable regulation of gene expression that do not involve a change in the DNA sequence. Epigenetic mechanisms include DNA methylation, histone modification such as acetylation, sumoylation, ubiquitination, and methylation, and non-coding RNAs.

In mammals, early embryogenesis is a critical period for the establishment of the epigenome [14]. Maternal obesity may influence the placenta and offspring epigenetic landscape [15,16]. Ge et al. found that DNA methylation in differential methylation regions of Peg3 is altered in spermatozoa of offspring from obese mothers, but is not affected in spermatozoa of offspring from diabetic mothers [17]. Epigenetic changes in the fetus as a consequence of in utero exposure to maternal obesity-related factors could have persistent effects and cause metabolic abnormalities at later ages.

### 3.2. Gut-Brain Axis

The bidirectional communication system, mediated by hormonal, immunological, and neural signals, between the gut and the brain constitutes the gut-brain axis. Pregnant mothers’ gut microbiota undergo dynamic compositional changes during gestation [18]. Inadequate nutrition during pregnancy has been related to altered maternal microbiota [19]. A human study showed that shifts in the gut microbiota during pregnancy are sensitive to the maternal pre-pregnancy BMI, as well as to weight gain during pregnancy [20]. In a prospective follow-up study of 256 women, infants’ fecal microbial composition was related to the weight and weight gain of their mothers during pregnancy. Infants of obese and high gestational weight gain women demonstrate lower concentrations of the genus *Bifidobacterium*, considered protective bacteria, and higher concentrations of proinflammatory bacteria, including *Bacterioides*, *Clostridium*, and *Staphylococcus* [21]. Similarly, in rodents, consumption of a high-fat diet (HFD) or a Western diet prior to and during pregnancy could alter the trajectory of maternal and offspring microbiota [22]. Putative mechanisms of dysbiosis that could lead to obesity might be due to increased energy extraction and storage from ingested nutrients [23].

### 3.3. Inflammation

Pregnancy itself involves a state of mild maternal systemic inflammation, with the placenta also actively produces a variety of immunomodulatory hormones and cytokines [24,25]. In humans, maternal systemic and placental inflammation has been observed in pregnancies complicated by obesity [26]. Maternal obesity leads to placental inflammation and increased cytokine production in human [27,28], rodent [29], and ovine models [30], including elevation of interleukin 6, interleukin 1β, and tumor necrosis factor-α production [28,30], as well as an increase in infiltrating monocytes and activated macrophages [27,28]. It has been shown that maternal inflammatory mediators during pregnancy correlate with fetal adiposity and neonatal fat mass [31]. Maternal obesity-mediated inflammation and placental-mediated inflammation may interact with each other, thus altering fetal development [32].

### 3.4. Mitochondrial Function

Mitochondria play an important role in metabolism and provide the principal energy source of the cell. Mitochondrial dysfunction has been identified in early embryogenesis in obese mothers. Igosheva et al. showed that maternal obesity prior to conception is associated with altered mitochondria in mouse oocytes and zygotes. Mouse oocytes and embryos of obese dams have increased mitochondrial membrane potential, higher levels of oxidative phosphorylation, and increased reactive oxygen species production compared to those of lean mothers. mtDNA copy number is also increased in response to oxidative damage [33]. 

### 3.5. Brain-Derived Neurotrophic Factor

Brain-derived neurotrophic factor (BDNF) is a target gene of miR-210. Maternal obesity could lead to increased levels of miRNA-210 and decreased levels of BDNF mRNA in placentas from female fetuses, and decreased proBDNF in placentas from male fetuses. In addition, maternal obesity adversely affects BDNF/tropomyosin receptor kinase B (TRKB) signaling in the placenta in a sexually dimorphic manner [34]. Recently, Fusco et al. showed maternal HFD-dependent insulin resistance downregulates BDNF and insulin signaling in maternal tissues and inhibits BDNF expression in both the germline and hippocampus of progeny [35]. The role of BDNF in developmental programming is highlighted by Briana et al. [36].

## 4. Programming Effects of Maternal Obesity Across Life-Span

### 4.1. Placental Alterations

The placenta is an important organ vital for fetal growth; compromised function is associated with abnormal fetus development. The link between in utero and later adult cardiovascular disease was first noticed by Barker et al. in an epidemiological study [37].

The placenta is both a target and producer of inflammatory mediators in the context of maternal obesity [25]. Maternal obesity leads to altered nutrient handling [38], energy modulation [39], and reduced angiogenesis. In humans, maternal obesity causes reduced chorionic plate artery function, which may reduce blood supply to the fetus and place the fetus at risk [40]. Rodent models of maternal obesity report decreased layer thickness, reduced trophoblast proliferation, increased macrophage activation, elevated cytokine gene expression, altered vascular development, and resultant hypoxia in the labyrinth [27,41]. 

### 4.2. Fetal Alterations

Fetuses of obese mothers are at increased risk of macrosomia and intrauterine growth retardation [38]. A previous study examined the fetal brain of term fetuses of obese rats fed a HFD. This study points to increased inflammation and oxidative stress in fetal brains of obese dams and dysregulation of monoamine neurotransmitter and hypothalamic orexigenic signaling. In immunohistochemical analysis, developmental alterations in the hypothalamic and extra-hypothalamic regions in the fetal brain from HFD dams are suggestive of a predisposition for the development of obesity and possibly neurodevelopmental abnormalities in offspring [42]. In a sheep study, fetal hearts from dams with maternal obesity display a normal cardiac contractile function during basal perfusion. However, they develop an impaired heart-rate-left-ventricular-developed pressure product following high workload stress [43]. Maternal obesity also affects adipose tissue development and differentiation during the fetal period [44,45].

### 4.3. Young Offspring Alterations

Infants born to overweight and obese mothers are more likely to be large for their gestational age and macrosomic [46]. Recently, a meta-analysis revealed a 264% increase in the odds of child obesity when mothers have obesity before conception [47]. In a rodent model, offspring from rats fed a diet with 50% fat calories from embryonic day 6 to postnatal day 15 are obese at 70 days of age [48]. During lactation, on postnatal day 15, the pups show an increase in the expression of 5 orexigenic peptides, both in the paraventricular nucleus (PVN) and in the parafornical lateral hypothalamus. Conversely, the expression of these orexigenic peptides is downregulated in the arcuate nucleus (ARC) [48]. The authors showed that perinatal HFD exposure reveals changes in the offspring hypothalamus that became apparent at preweaning [48]. Of note, other factors such as maternal education, socioeconomic position, early childhood environment and nutrition are also crucial determinants of early childhood development [49].

### 4.4. Adult Offspring Alterations

Epidemiological and experimental studies show that maternal obesity is associated with increased susceptibility to hepatic steatosis and inflammation [50], hypertension and cardiovascular disorders [51], kidney disease [52], neurodevelopmental disorders [42] as well as obesity and insulin resistance [53] in offspring at different stages of development. Below, we summarize the programming effects in a few main organs in the context of maternal obesity. 

#### 4.4.1. Brain

Mounting evidence from both prenatal and lactational human epidemiologic and animal studies suggests that exposure to maternal obesity and a HFD are associated with neurodevelopmental and psychiatric disorders in offspring. These disorders include cognitive impairment, autism spectrum disorders, attention deficit hyperactivity disorder, mood disorders, schizophrenia, and eating disorders [11].

The hypothalamus plays a critical role in energy homeostasis by regulating both appetite and energy expenditure. Hypothalamic ARC is a putative site of appetite regulation and translates peripheral signals to orexigenic agouti-related peptide (AgRP) or anorexigenic pro-opiomelanocortin (POMC)-expressing neurons. Desai et al. showed that adult rat offspring born to high-fat fed dams are obese at six months of age. They demonstrated that maternal obesity programs the offspring hypothalamic ARC, resulting in enhanced activity in AgRP neurons relative to POMC (pro-opiomelanocortin) neurons and hyperphagia [54]. Prior studies also showed maternal obesity/overnutrition programs offspring hyperphagia via appetite/satiety gene expression [48,55].

The hippocampus is critical to learning and memory and is especially susceptible to early-life nutritional insults. Tozuka et al. reported that mouse pups from dams fed with a HFD show peroxidized lipid accumulation in many brain regions, impaired adult neurogenesis in the hippocampus, and deficits in spatial learning performance [56,57]. Impaired hippocampal BDNF expression in the context of maternal obesity has been linked to deficits in spatial learning and memory in juvenile and adult offspring [57,58]. It is conceivable that the deleterious effects of maternal obesity on offspring learning and memory may be mediated by alterations of BDNF-mediated synaptic plasticity.

#### 4.4.2. Liver

Animal models have demonstrated that offspring of diet-induced obese dams develop metabolic complications, including nonalcoholic-fatty liver disease (NAFLD) [50]. Offspring of maternal obesity have increased liver weight and lipid accumulation. Additionally, upregulated sterol regulatory element-binding protein 1 and 20 downstream genes involved in hepatic lipid biosynthesis and reduced expression of peroxisome proliferator-activated receptor alpha (PPARα) and related genes involved in fatty acid oxidation are involved in the pathogenesis of NAFLD in the context of maternal obesity [59]. 

In a mouse model of maternal obesity, offspring develop NAFLD prior to any differences in body weight or body composition. Offspring of obese dams have higher liver lipid content, as well as increased levels of peroxisome proliferator-activated receptor gamma (PPARγ) and reduced levels of triglyceride lipase. Increased liver glycogen and protein content are also observed in offspring of obese dams [50].

#### 4.4.3. Kidney

Armitage et al. intended to obtain better insight into hypertension in offspring from maternally obese dams. They found renal stereology showed no differences in kidney weight, glomerular number, or volume in adult rat offspring from dams fed a lard-rich diet compared with dams fed a control diet. However, renal renin and Na^+^, K^+^-ATPase activity are significantly reduced in adult offspring of maternally obese dams compared with controls [60].

In a mouse model of maternal obesity, the renal function of offspring from obese mothers was considerably worse compared to offspring from non-obese mothers, especially when they developed diabetes. Glastras et al. found that adult male mice offspring of obese mothers have increased renal fibrosis, inflammation, and oxidative stress. Furthermore, offspring exposed to maternal obesity have increased susceptibility to renal damage when a second hit of streptozotocin is imposed. The authors proposed that maternal obesity should be considered a risk factor for chronic kidney disease [52].

#### 4.4.4. Cardiovascular System

The increased rates of hypertension in modern society may have resulted from the nutritional environment during early life. A population-based birth cohort of 2432 Australians showed that greater maternal gestational weight gain is independent from maternal pre-pregnancy body mass index, associates with a higher body mass index and tends to be associated with a higher systolic blood pressure in the offspring at age 21 of years [61]. Armitage et al. studied hypertension in offspring of maternally obese mothers. Using aorta stereology, they show increased aortic stiffness and reduced endothelium-dependent relaxation in adult offspring of maternally obese dams compared with controls [60]. Recently, a study using birth records from 37,709 participants showed that a higher maternal body mass index at the first antenatal visit is associated with an increased risk of premature death and hospital admissions for cardiovascular events in adult offspring. The offspring of overweight mothers also carried a higher risk of adverse outcomes compared to offspring of mothers with a normal BMI [62]. 

Maternal obesity and/or HFD exposure may predispose offspring to a higher prevalence of postnatal fat diet-induced metabolic diseases including cardiovascular complications later on in adult life [9].

#### 4.4.5. Adipose Tissue

Maternal obesity affects adipocyte development and alters functional properties of adipocytes from the fetal period [44] to adulthood [63], resulting in higher white adipose tissue (WAT) mass and larger adipocytes [64]. Upregulated PPARγ, whose activity is enhanced under limited or excess nutrient availability, is one of the important features of enhanced adipogenesis and fat expansion in programmed offspring of obese dams [44]. This is associated with downregulation of PPARγ corepressors [65]. 

## 5. Reprogramming the Adversities Encountered in Maternal Obesity

Reprogramming strategies refers to maneuvers to reverse malprogrammed development to resume normal development [66,67]. These strategies could be nutritional interventions, lifestyle modifications, or pharmacological interventions [68]. A mechanistic understanding of how the maternal intrauterine and lactational environment may be mediating offspring common morbidities is critical, because early life may represent an important window for targeted intervention to ameliorate fetal and offspring risk [66]. A previous review suggests that early-life resveratrol supplementation could be considered as a reprogramming strategy against the development of metabolic syndrome-related disorders [67]. Supplementing with certain classes of nutrients in gestation is helpful in reprogramming of maternal adversities. Resveratrol has powerful anti-oxidant and anti-inflammatory activities. Resveratrol can cross placenta and seems to have no obvious toxicity in humans. Below, we discuss the safety profile of resveratrol taken during pregnancy and its possible reprogramming potential in the context of maternal obesity. Figure 1 depicts the programming effects of maternal obesity and the reprogramming effects of maternal resveratrol intake. 

## 6. Resveratrol Safety Profiles

Resveratrol (trans-3,5,4’-trihydroxystilbene) is a phenolic compound with powerful anti-oxidant activity, which is found in various plants such as grapes and berries [69]. In humans, resveratrol has been reported to be safe at doses up to 5 g per day [70]; however, trans-resveratrol at a dose of 2000 mg twice daily resulted in diarrhea and clinically irrelevant changes of serum potassium and total bilirubin concentrations in healthy volunteers [71].

In a comprehensive study of the safety profile of resveratrol, a 28-day study of resveratrol causes no adverse effects in rats at 50, 150, and 500 mg/kg/day after 28 days of gavage intake. In addition, resveratrol did not cause any adverse effects in rats at up to 700 mg/kg/day after 90 days. In terms of reproductive toxicity assays, 750 mg/kg/day of resveratrol did not induce any adverse reproductive effects in an embryo-fetal study [72]. 

## 7. Maternal Resveratrol Administration in the Context of Maternal Adversities

Maternal resveratrol intake has been shown to alleviate impaired hippocampal neurogenesis and increase hippocampal BDNF expression in offspring of prenatally stressed dams [73]. Similarly, maternal resveratrol supplementation prevents prenatal stress-induced cognitive spatial deficits accompanied by enhanced hippocampal mitochondrial function and reduced oxidative stress [74]. Care et al. showed that maternal resveratrol supplementation during pregnancy and lactation ameliorated the development of hypertension in adult offspring [75]. Likewise, Chen et al. reported that maternal resveratrol intake during pregnancy and lactation could alleviate programmed hypertension induced by combined maternal N^G^-nitro-L-arginine-methyl ester treatment and postnatal HFD [76]. In preeclampsia patients, both the time and dose of blood pressure control are significantly reduced in those who received resveratrol supplementation [77]. Recently, Ding et al. scrutinized the potential use of resveratrol in adverse human pregnancies [78].

## 8. Possible Beneficial Mechanisms of Maternal Resveratrol Administration in the Context of Maternal Obesity

The beneficial effects of maternal intake of resveratrol on reprogramming might be via its actions on the pregnant mother, placenta, or fetus. Resveratrol could influence placental function through its anti-inflammatory [79] and anti-oxidant activities [80]. From the perspective of the fetus, resveratrol can cross the placenta and affect the fetus directly [81]. From the maternal perspective, resveratrol has been shown to protect against HFD-induced obesity in animal studies [82,83] and exerts health benefits in obese individuals [84]. 

It was reported that resveratrol is able to modulate methylation and acetylation of lysine 9 of histone H3 in zygotic pronuclei [85]. These changes in zygotes may lead to more successful preimplantation embryo development. Likewise, pregnant mice supplemented with quercetin, a flavonoid, show significantly enhanced expression of Cyp1a1, Cyp1b1, Nqo1, and Ugt1a6 in mouse fetal liver tissues at gestational day 14.5, which persisted into adulthood in a tissue- and gender-dependent manner [86].

Resveratrol could affect gut microbiota and their metabolic products, such as short chain fatty acids and intraluminal lipids, and hence is helpful in the context of obesity and its associated conditions [87,88].

Anti-obesity activity of resveratrol has been reported in laboratory animal models, including mice [89], rats [90], primates [91], and humans [92]. The attributed pathways occur in the WAT and include adipogenesis, lipolysis, and lipogenesis [93]. Resveratrol may act as a fat browning activator, leading to enhanced glucose homeostasis and weight loss, and involvement in the secretion of many myokines and adipokines [82]. It is conceivable that resveratrol might reduce maternal obesity-mediated inflammation.

Resveratrol can modulate mitochondrial ultra-structure, function, and dynamics, and cause alterations in mitochondrial physiology [94]. Resveratrol has beneficial effects in mitochondrial function in an ex vivo study performed in muscle fibers from HFD mice. Resveratrol induces PGC-1α activity by facilitating sirtuin 1-mediated deacetylation [89]. It is cautioned that resveratrol may cause mitotoxicity in the human placental explant [95].

Resveratrol may have antidepressant-like effects in stressed rats, mediated in part by the up-regulation of BDNF levels in the hippocampus and amygdala [96]. Maternal BDNF may reach the fetal brain through the utero-placental barrier [97]. To the best of our knowledge, there are no reports in the literature on the effects of resveratrol on the embryonic expression of BDNF. 

Current trends include improved management during pregnancy, such as approaches to reduce BMI before conception in women of reproductive age and increased attention to postpartum weight management and vigorous maternal glycemic control. A behavioral intervention with diet and physical activity in obese mothers is insufficient to reduce the incidence of fetal macrosomia or to prevent the occurrence of gestational diabetes mellitus [98]. Since maternal obesity can induce negative influences in offspring beginning from the embryo through development, the potential implication of maternal resveratrol administration in reprogramming is worth investigation [99]. 

In Grove’s laboratory, they studied pregnant Japanese macaques fed a Western-style diet (36% fat) supplemented with 0.37% resveratrol throughout pregnancy. Resveratrol use during pregnancy yields improvements in the maternal and placental phenotype with beneficial effects in the fetal liver. However, the fetal pancreatic mass was enlarged by 42%, with a 12-fold increase in Ki67 immunohistochemistry in the exocrine compartment of the pancreas. Thus, the authors cautioned against the use of resveratrol by pregnant women [100]. Additionally, the same authors found that consumption of a Western-style diet resulted in impaired fetal islet capillary density and sympathetic islet innervation, suggesting a novel mechanism by which maternal Western-style diet consumption leads to increased susceptibility to type 2 diabetes in offspring. Resveratrol mitigated maternal diet-induced harmful effects in the pancreas in the fetus and juvenile offspring. However, resveratrol supplementation caused pancreas hypervascularization compared with controls. Thus, the authors again stated that the use of resveratrol in pregnancy required caution [101].

In a mouse model, Zou et al. found that maternal HFD intake during pregnancy and lactation impairs the development of offspring brown adipose tissue (BAT) and beige adipocytes at weaning and has persistent effects on the metabolic health of adult offspring. Importantly, administration of 0.2% (W/W) maternal resveratrol promotes a thermogenic program in BAT and WAT during early offspring development, and induces beige adipocyte development of WAT in adult male offspring. In addition, maternal resveratrol intervention protects offspring against HFD-induced obesity [102].

Ros et al. administered resveratrol in the drinking water (50 mg/L) of Wistar rats fed a high-fat diet (fat; 61.6%) during pregnancy and lactation. Resveratrol did not affect the body weight, visceral fat mass, or serum leptin levels in pregnant/lactating dams. Resveratrol intake during pregnancy and lactation decreases body weight, adipose tissue, lipid profiles, and serum leptin levels at weaning in offspring from dams on a HFD [103].

One clinical study performed in overweight pregnant women found that 80 mg of resveratrol shows significantly improved the lipid and glucose parameters compared to controls after 30 or 60 days of treatment [104]. Table 1 summarizes the current available clinical and animal studies performed with maternal administration of resveratrol for the intervention of maternal obesity/overweight.

## 9. Conclusions

Public health approaches to reduce obesity epidemic is a top priority [105]. The transmission of the altered phenotypes to the subsequent generations in the context of maternal obesity should not be overlooked. Hence, this seems to be one of the very important tasks of preventive medicine in the current practice.

The future tasks will be to translate this mechanistic understanding into prevention and directed interventions aimed at reducing the burden of maternal obesity. Approaches to reduce BMI before conception in women of reproductive age and increased attention to postpartum weight management and vigorous maternal glycemic control are encouraged. Resveratrol has a lower intestinal absorption rate and poor solubility. Therefore, the future task is the development of a resveratrol formulation with better pharmacokinetic and pharmacodynamic properties. Clinical trials aiming to determine the ideal dosage and therapeutic window of resveratrol as the reprogramming therapy are needed. In addition, the potential role of resveratrol in reprogramming the adversities of maternal obesity worth further research.

## Figures and Tables

**Figure 1 ijerph-17-01610-f001:**
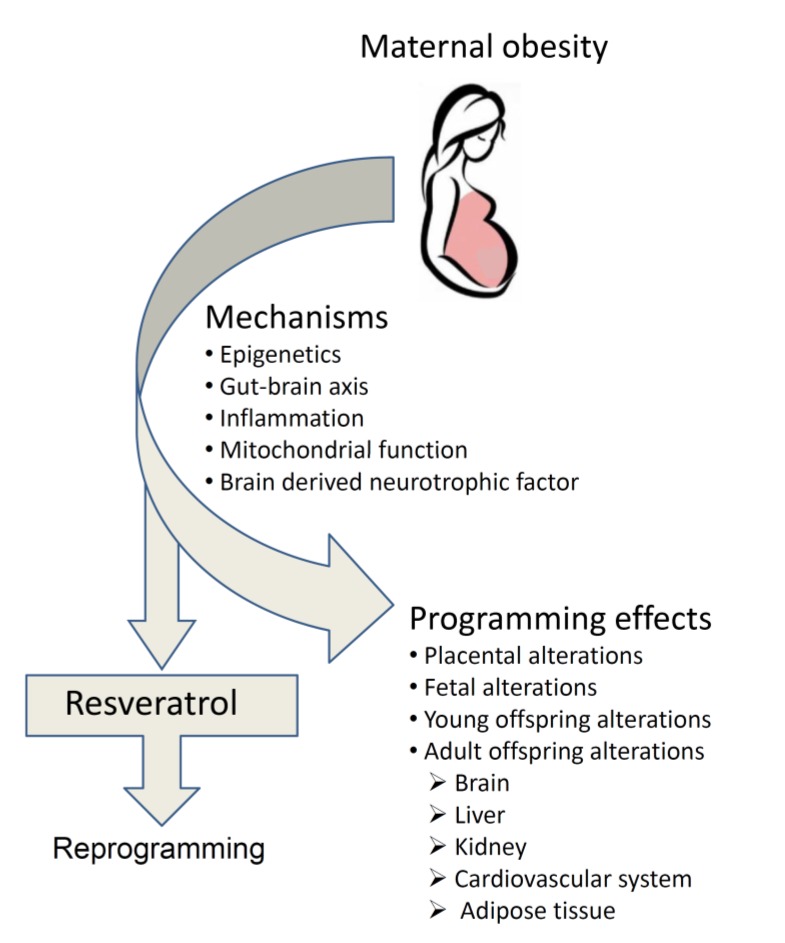
Programming effects and underlying mechanisms of maternal obesity and reprogramming effects of maternal resveratrol intake.

**Table 1 ijerph-17-01610-t001:** Clinical and animal studies of maternal obesity/overweight treated with maternal resveratrol.

Gender/Species	Diet Module Obesity	Dose and Period of Resveratrol Supplementation	Maternal Obesity/Over-Weight Offspring Obesity Group Size	Age at Evaluation Group Size	Major Beneficial Findings in Offspring	Reference
Japanese macaques	Maternal Western-style (36% fat) diet	0.37% *w*/*w* resveratrol in diet from 3 months before breeding and until gestational day 130	+/−N = 6	Gestational day 130	Restored the loss of fetal islet vascularization	[101]
Japanese macaques	Maternal Western-style (36% fat) diet	0.37% *w*/*w* resveratrol in diet from 3 months before breeding and until gestational day 130	+/−N = 6	Gestational day 130	30% maternal weight loss, increased uterine artery blood flow, decreased placenta inflammation, reduced fetal liver triglyceride deposition	[100]
Male and female Wistar rats	Maternal high-fat diet (61.6%)	Resveratrol (50 mg/L) in drinking water during pregnancy and lactation	+/+N = 4–6	3 weeks	Attenuated hyperglycemia, obesity and hyperlipidemia. Resveratrol reduced body weight, leptin, visceral adipose tissue, and subcutaneous adipose tissue, with females being more affected	[102]
Japanese macaques	Maternal Western-style (36% fat) diet	0.37% *w*/*w* resveratrol in diet from 3 months before breeding and until gestational day 130	+/− N = 6	Gestational day 130 N = 5–10	Resveratrol stimulated placental DHA uptake activity, AMPK activation, and transporter expression	[95]
Male C57BL/6 J mice	Maternal plus postnatal high-fat diet for 11 weeks	0.2% *w*/*w* (~200 mg/kg/day) resveratrol in diet during pregnancy and lactation	+/+ N = 10	14 weeks	Promotes beige adipocyte development in offspring white adipose tissue. Protects offspring against high-fat diet-induced obesity	[101]
Italy	Human	Resveratrol 80mg/day	+/N 110 pregnant women	30 days and 60 days	Supplementation of resveratrol to DCI/MI improves mother glucose and lipid control	[103]

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
