# Peer review of "Maternal Obesity Programs Offspring Development and Resveratrol Potentially Reprograms the Effects of Maternal Obesity"

_ijerph, 2020, doi:10.3390/ijerph17051610_

Round 1
Reviewer 1 Report
The manuscript reviewed the influence of maternal obesity on programming of offspring health, as well as the potential benefits of resveratrol on preventing adverse effect of maternal obesity. The following questions need to be addressed:
The manuscript contains several typos and grammatical errors, affecting readability of the manuscript. Please proofread the manuscript carefully and if possible, have a native English speaker edit the manuscript. It seems that the content of the manuscript is divided into two sections, the influence of maternal obesity on prenatal programming, and the influence of resveratrol on prenatal programming. Each section is brief and lacks in depth. The two sections are also not well connected, e.g. are the mechanisms/functional changes due to maternal obesity mentioned in section 1 addressed by resveratrol treatment in section 2? These are of the most interest to the readers and should be more extensively discussed. Since the topic of maternal obesity on prenatal programming has been reviewed in many other publications, it would be more innovative to focus on resveratrol and its influence on prenatal programming.Author Response
The manuscript contains several typos and grammatical errors, affecting readability of the manuscript. Please proofread the manuscript carefully and if possible, have a native English speaker edit the manuscript. It seems that the content of the manuscript is divided into two sections, the influence of maternal obesity on prenatal programming, and the influence of resveratrol on prenatal programming. Each section is brief and lacks in depth. The two sections are also not well connected, e.g. are the mechanisms/functional changes due to maternal obesity mentioned in section 1 addressed by resveratrol treatment in section 2? These are of the most interest to the readers and should be more extensively discussed. Since the topic of maternal obesity on prenatal programming has been reviewed in many other publications, it would be more innovative to focus on resveratrol and its influence on prenatal programming.
Reply: thanks for your point.
This manuscript has been sent for editorial assistance by a native English speaker. (Editage Taiwan)
Reply: thanks for your point.
Since there were not many publications regarding resveratrol on maternal obesity, therefore, we tried our best to match section 3 (section 1) and 8 (section 2), although not a point-to point connection.
We adopt other reviewers’ advices to make this article more balanced in viewpoints and comprehensive in perspectives.
Reply: thanks for your point.
We add some relevant discussion of resveratrol on prenatal programming.
Line 278:
Maternal resveratrol intake has been shown to alleviate impaired hippocampal neurogenesis and increase hippocampal BDNF expression in offspring of prenatal stress dams [70]. Similarly, resveratrol maternal supplementation prevents prenatal stress induced cognitive spatial deficit accompanied by enhancing hippocampal mitochondrial function and reduced oxidative stress [71]. Care et al. showed that maternal resveratrol supplementation during pregnancy and lactation ameliorated the development of hypertension in adult offspring [75]. Likewise, Chen et al. reported that maternal resveratrol intake during pregnancy and lactation could alleviate programmed hypertension induced by combined maternal NG-nitro-L-arginine-methyl ester treatment and postnatal HFD [76]. In preeclampsia patients, both the time and doses of blood pressure control were significantly reduced in who received resveratrol supplementation [77]. Recently, the potential use of resveratrol in adverse human pregnancies is under scrutinization [78].
Overall an interesting and helpful overview of the mechanistic effects of maternal obesity on offspring health. The authors through this article explored the mechanisms underlying the programming effects of maternal obesity. In addition, the evidence on the effectiveness of resveratrol to prevent the effects of maternal obesity has been reviewed.
Abstract:
The abstract repeats certain points and could potentially include some key findings of the review instead. Line 12 unclear “culprits”. Line 14-15 – Do you mean the effect on child’s risk of future obesity, or the mothers' risk of obesity post-partum?
Reply:
We re-wording as following:
Maternal obesity during pregnancy is a now a public health burden that may be the culprit underlying the ever-increasing rates of adult obesity worldwide. (line 12)
Understanding the association between maternal obesity and adult offspring’s obesity would inform policy and practice regarding offspring health through available resources and interventions. (line 13)
Reviewer 2 Report
Summary:
Overall an interesting and helpful overview of the mechanistic effects of maternal obesity on offspring health. The authors through this article explored the mechanisms underlying the programming effects of maternal obesity. In addition, the evidence on the effectiveness of resveratrol to prevent the effects of maternal obesity has been reviewed.
Abstract:
The abstract repeats certain points and could potentially include some key findings of the review instead. Line 12 unclear “culprits”. Line 14-15 – Do you mean the effect on child’s risk of future obesity, or the mothers' risk of obesity post-partum?
Introduction:
The language used while describing evidence is quite strong – e.g. Line 49 “accepted fact”. This can perhaps be rephrased to “growing evidence suggests that.” etc. This was noted through the paper, please note that many of these studies showed associations that did not establish causality.
Maternal obesity is strongly linked with socioeconomic status and environmental determinants of health (and others e.g. endocrine disruptors). I understand that reviewing and adding details on the social and policy-related causes of maternal obesity is beyond the scope of this article. Perhaps it can be acknowledged towards the end, as factors that cannot be overlooked? This may be of interest to readers of the international journal of Environmental Research and Public Health.
Methods/ Structure: As it is a review article detailed methods are not mentioned, but the authors could add a note on which databases were explored (if conducted for this study).
The article is structured well with subheadings that are clear and helpful. However, there could be a section (for example a short discussion section) that links together the points made in the article in the context of existing evidence and risks of obesity. E.g. what are the recommendations by the authors based on the review (some more insights could be helpful as the conclusion is very short)? For example, would the authors recommend preconception weight control over interventions in pregnancy, since the literature review shows pre-pregnancy BMI as a strong risk factor?
Lines 279-283: We must also be careful about the issue of over-medicalization of pregnancies. Resveratrol may help (currently inadequate human studies), but other risks related to the mother’s weight gain needs to be addressed through lifestyle interventions for optional nutrition and physical activity. Though some interventions have not helped improve all fetal outcomes, the number of RCTs looking at various behavioural and nutritional interventions are increasing with promising findings, and this must not be dismissed, (e.g. Poston et al referenced here was effective in some of the outcomes such as reducing dietary glycaemic load, improved maternal dietary habits and physical activity, and gestational weight gain). (please see suggested references below, and comment on the introduction section about how obesity is also a socio-cultural issue).
The role of diabetes (GDM and type 2) as a mediating factor in worsening fetal outcomes and increased maternal risk of NCDs in later life is also important to be mentioned.
Section 4.4.1
While maternal obesity is associated with the outcomes mentioned here, please note that other factors such as maternal education, socioeconomic status, early childhood environment and nutrition are crucial determinants of early childhood development (Britto et al Lancet)
Line 222-224 – which outcomes were considered in this study?
Conclusion:
This section is very short. Some specific options on how maternal obesity and the transgenerational passage of risk can be prevented could be added here (if limited by word count for a discussion section). The suggested references are based on my comments listed below.
Other comments:
Thorough proofreading is required as errors in grammar were seen.
e.g. Line 41-43 – needs re-wording
Sectional headings 4.1 and 4.2 could be placental and fetal alterations respectively.
Suggested references:
Britto, Pia R., et al. "Nurturing care: promoting early childhood development." The Lancet 389.10064 (2017): 91-102.
Hanson, Mark, et al. "Interventions to prevent maternal obesity before conception, during pregnancy, and post-partum." The lancet Diabetes & endocrinology 5.1 (2017): 65-76.
Ma, Ronald Ching Wan, et al. "Clinical management of pregnancy in the obese mother: before conception, during pregnancy, and post-partum." The lancet Diabetes & endocrinology 4.12 (2016): 1037-1049.
Grobler L, Visser M, Siegfried N. Healthy Life Trajectories Initiative: Summary of the evidence base for pregnancy‐related interventions to prevent overweight and obesity in children. Obesity Reviews. 2019 Aug;20:18-30.
Author Response
Introduction:
The language used while describing evidence is quite strong – e.g. Line 49 “accepted fact”. This can perhaps be rephrased to “growing evidence suggests that.” etc. This was noted through the paper, please note that many of these studies showed associations that did not establish causality.
Reply: We re-wording as following:
Growing evidence suggests that developmental programming is noted in obese mothers’ offspring [13].
Maternal obesity is strongly linked with socioeconomic status and environmental determinants of health (and others e.g. endocrine disruptors). I understand that reviewing and adding details on the social and policy-related causes of maternal obesity is beyond the scope of this article. Perhaps it can be acknowledged towards the end, as factors that cannot be overlooked? This may be of interest to readers of the international journal of Environmental Research and Public Health.
Reply: thanks.
We adopt your advice and add one sentence and reference in introduction.
Line 30:
Maternal obesity is also linked with individual’s socioeconomic position and environmental determinants of health [6].
Bann D, Cooper R, Wills AK, Adams J, Kuh D; NSHD scientific and data collection team. Socioeconomic position across life and body composition in early old age: findings from a British birth cohort study. J Epidemiol Community Health. 2014 Jun;68(6):516-23. doi:10.1136/jech-2013-203373.
Methods/ Structure: As it is a review article detailed methods are not mentioned, but the authors could add a note on which databases were explored (if conducted for this study).
Reply: thanks.
We added in Introduction:
In line 36:
The works cited in this review were identified eligible studies and reviews through manual searches related to maternal obesity.
The article is structured well with subheadings that are clear and helpful. However, there could be a section (for example a short discussion section) that links together the points made in the article in the context of existing evidence and risks of obesity. E.g. what are the recommendations by the authors based on the review (some more insights could be helpful as the conclusion is very short)? For example, would the authors recommend preconception weight control over interventions in pregnancy, since the literature review shows pre-pregnancy BMI as a strong risk factor?
Reply: thanks.
We add one paragraph in Discussion conclusion.
Lines 283-287: We must also be careful about the issue of over-medicalization of pregnancies. Resveratrol may help (currently inadequate human studies), but other risks related to the mother’s weight gain needs to be addressed through lifestyle interventions for optional nutrition and physical activity. Though some interventions have not helped improve all fetal outcomes, the number of RCTs looking at various behavioural and nutritional interventions are increasing with promising findings, and this must not be dismissed, (e.g. Poston et al referenced here was effective in some of the outcomes such as reducing dietary glycaemic load, improved maternal dietary habits and physical activity, and gestational weight gain). (please see suggested references below, and comment on the introduction section about how obesity is also a socio-cultural issue).
Reply: thanks.
We add some points to balance this notion. Line 322:
Current trends include improved management during pregnancy, such as approaches to reduce BMI before conception in women of reproductive age and increased attention to postpartum weight management and vigorous maternal glycemic control [98].
Also, possible side effect of maternal resveratrol was mentioned.
Line: 340
However, resveratrol supplementation caused pancreas hypervascularization compared with controls. They stated that the use of resveratrol in pregnancy needed further caution [96].
The role of diabetes (GDM and type 2) as a mediating factor in worsening fetal outcomes and increased maternal risk of NCDs in later life is also important to be mentioned.
Reply: thanks. We add in line 48
The rise in obesity prevalence is partly due to increase in pre-existing type 2 diabetes among pregnant women. Besides, maternal obesity is strongly linked with gestational diabetes [12].
Marchi J, Berg M, Dencker A, Olander EK, Begley C. Risks associated with obesity in pregnancy, for the mother and baby: a systematic review of reviews. Obes Rev. 2015 Aug;16(8):621-38. doi: 10.1111/obr.
Section 4.4.1
While maternal obesity is associated with the outcomes mentioned here, please note that other factors such as maternal education, socioeconomic status, early childhood environment and nutrition are crucial determinants of early childhood development (Britto et al Lancet)
Reply: we add Britto et al.’s work in section
In line 145:
Of note, other factors such as maternal education, socioeconomic position, early childhood environment and nutrition are also crucial determinants of early childhood development [49].
Line 222-224 – which outcomes were considered in this study?
Reply: we re-wording as following:
A previous review suggests that early-life resveratrol supplementation could be considered as a reprogramming strategy against the development of metabolic syndrome-related disorders [67].
Conclusion:
This section is very short. Some specific options on how maternal obesity and the transgenerational passage of risk can be prevented could be added here (if limited by word count for a discussion section). The suggested references are based on my comments listed below.
Reply:
We re-write the Conclusion as following:
Public health approaches to reduce obesity epidemic is a top priority [Hanson; 27743974]. The transmission of the altered phenotypes to the subsequent generations in the context of maternal obesity should not be overlooked. Hence, this seems to be one of the very important tasks of preventive medicine in the current practice.
The future tasks will be to translate this mechanistic understanding into prevention and directed interventions aimed at reducing the burden of maternal obesity. Approaches to reduce BMI before conception in women of reproductive age and increased attention to postpartum weight management and vigorous maternal glycemic control are encouraged. In addition, the potential role of resveratrol in reprogramming the adversities of maternal obesity worth further research.
Other comments:
Thorough proofreading is required as errors in grammar were seen.
e.g. Line 41-43 – needs re-wording
Reply: This manuscript has been sent for editorial assistance by a native English speaker. (Editage Taiwan)
We re-wording as following:
Clinical and animal studies have shown a maternal obesogenic environment, i.e., during the periods of periconception, pregnancy, or lactation, increasing the risk for the development of obesity and related cardiometabolic [9, 10] and neurodevelopmental disorders [11] in offspring in adulthood.
Sectional headings 4.1 and 4.2 could be placental and fetal alterations respectively.
Reply: thanks.
Changed accordingly.
Suggested references:
Britto, Pia R., et al. "Nurturing care: promoting early childhood development." The Lancet 389.10064 (2017): 91-102.
Hanson, Mark, et al. "Interventions to prevent maternal obesity before conception, during pregnancy, and post-partum." The lancet Diabetes & endocrinology 5.1 (2017): 65-76.
Ma, Ronald Ching Wan, et al. "Clinical management of pregnancy in the obese mother: before conception, during pregnancy, and post-partum." The lancet Diabetes & endocrinology 4.12 (2016): 1037-1049.
Grobler L, Visser M, Siegfried N. Healthy Life Trajectories Initiative: Summary of the evidence base for pregnancy‐related interventions to prevent overweight and obesity in children. Obesity Reviews. 2019 Aug;20:18-30.
Reply:
Suggested references were added (49 and 105).
Reviewer 3 Report
The manuscript focuses on one of the most important public health problems, i.e. obesity. The study is mainly concentrated on associations between maternal obesity and intrauterine programming of offspring. The second shorter part of the manuscript refers to potential reprogramming effects of resveratrol administered to obese pregnant mothers. This point of view seems to be very interesting.
However, there are some limitations:
The authors categorized two different conditions ("pre-pregnancy obesity" and "excessive gestational weight gain") into one term, that is "maternal obesity". The severity and types of metabolic disturbances in pre-pregnancy maternal obesity and excessive gestational weight gain should not be equated.
There are some abbreviations in the manuscript, which should be explained.
Please use "Instruction for Authors" for all references.
Author Response
The manuscript focuses on one of the most important public health problems, i.e. obesity. The study is mainly concentrated on associations between maternal obesity and intrauterine programming of offspring. The second shorter part of the manuscript refers to potential reprogramming effects of resveratrol administered to obese pregnant mothers. This point of view seems to be very interesting.
However, there are some limitations:
The authors categorized two different conditions ("pre-pregnancy obesity" and "excessive gestational weight gain") into one term, that is "maternal obesity". The severity and types of metabolic disturbances in pre-pregnancy maternal obesity and excessive gestational weight gain should not be equated.
Reply:
Thanks for your point. We rephrase as following:
In line 38
The works cited in this review were identified eligible studies and reviews through manual searches related to maternal obesity. It is well known that pre-pregnancy maternal obesity and excessive gestational weight gain have different mechanisms and clinical implications. In this manuscript, we use the term maternal obesity to mean maternal obesity during pregnancy.
There are some abbreviations in the manuscript, which should be explained.
Reply: explained.
Please use "Instruction for Authors" for all references.
Reply: corrected.
Round 2
Reviewer 1 Report
- Line 205, section 5, please consider addressing why resveratrol is a particularly interesting/appropriate reprogramming strategy (as compared with other many nutrients/bioactive components in the diet)
2. Section 9. 333, please summarize the current status regarding research on the reprogramming effect of resveratrol and pinpoint some future directions of research regarding this research area.
Author Response
1. Line 205, section 5, please consider addressing why resveratrol is a particularly interesting/appropriate reprogramming strategy (as compared with other many nutrients/bioactive components in the diet)
Response: red-color text was added
Supplementing with certain classes of nutrients in gestation is helpful in reprogramming of maternal adversities. Resveratrol has powerful anti-oxidant and anti-inflammatory activities. Resveratrol can cross placenta and seems to have no obvious toxicity in humans. Below, we discuss the safety profile of resveratrol taken during pregnancy and its possible reprogramming potential in the context of maternal obesity.
2. Section 9. 333, please summarize the current status regarding research on the reprogramming effect of resveratrol and pinpoint some future directions of research regarding this research area.
Response: red-color text was added
The future tasks will be to translate this mechanistic understanding into prevention and directed interventions aimed at reducing the burden of maternal obesity. Approaches to reduce BMI before conception in women of reproductive age and increased attention to postpartum weight management and vigorous maternal glycemic control are encouraged. Resveratrol has a lower intestinal absorption rate and poor solubility. Therefore, the future task is the development of a resveratrol formulation with better pharmacokinetic and pharmacodynamic properties. Clinical trials aiming to determine the ideal dosage and therapeutic window of resveratrol as the reprogramming therapy are needed. In addition, the potential role of resveratrol in reprogramming the adversities of maternal obesity worth further research.